# Major acute cardiovascular events after dengue infection–A population-based observational study

Kai-Che Wei[1,2], Cheng-Len Sy[3], Wen-Hwa Wang[4,5,6], Chia-Ling Wu[7], Shang-Hung Chang[7,8,9]*, Yu-Tung Huang[7]*

1 Department of Dermatology, Kaohsiung Veterans General Hospital, Kaohsiung, Taiwan, 2 School of Medicine, College of Medicine, National Yang Ming Chiao Tung University, Taipei, Taiwan, 3 Department of Internal Medicine, Division of Infectious Diseases, Kaohsiung Veterans General Hospital, Kaohsiung, Taiwan, 4 Department of Internal Medicine, Division of Cardiology, Kaohsiung Veterans General Hospital, Kaohsiung, Taiwan, 5 Health Management Center, Kaohsiung Veterans General Hospital, Kaohsiung, Taiwan, 6 College of Management, I-Shou University, Kaohsiung, Taiwan, 7 Department of Medical Research and Development, Center for Big Data Analytics and Statistics, Chang Gung Memorial Hospital Linkou Main Branch, Taoyuan, Taiwan, 8 Department of Internal Medicine, Division of Cardiology, Chang Gung Memorial Hospital Linkou Main Branch, Taoyuan, Taiwan, 9 College of Medicine, Chang Gung University, Taoyuan, Taiwan

* afen.chang@gmail.com (SH-C); anton.huang@gmail.com (YT-H)

**Data Availability Statement:** Data cannot be shared publicly because of the policy by HWDC, MoHW based on the Personal Data Protection Act. Contact information for data application: https://

## Abstract

### Background

Dengue virus (DENV) infection may be associated with increased risks of major adverse cardiovascular effect (MACE), but a large-scale study evaluating the association between DENV infection and MACEs is still lacking.

### Methods and findings

All laboratory confirmed dengue cases in Taiwan during 2009 and 2015 were included by CDC notifiable database. The self-controlled case-series design was used to evaluate the association between DENV infection and MACE (including acute myocardial infarction [AMI], heart failure and stroke). The "risk interval" was defined as the first 7 days after the diagnosis of DENV infection and the "control interval" as 1 year before and 1 year after the risk interval. The incidence rate ratio (IRR) and 95% confidence interval (CI) for MACE were estimated by conditional Poisson regression. Finally, the primary outcome of the incidence of MACEs within one year of dengue was observed in 1,247 patients. The IRR of MACEs was 17.9 (95% CI 15.80–20.37) during the first week after the onset of DENV infection observed from 1,244 eligible patients. IRR were significantly higher for hemorrhagic stroke (10.9, 95% CI 6.80–17.49), ischemic stroke (15.56, 95% CI 12.44–19.47), AMI (13.53, 95% CI 10.13–18.06), and heart failure (27.24, 95% CI 22.67–32.73). No increased IRR was observed after day 14.

### Conclusions

The risks for MACEs are significantly higher in the immediate time period after dengue infection. Since dengue infection is potentially preventable by early recognition and vaccination,

dep.mohw.gov.tw/dos/cp-5119-59201-113.html
All databases were encrypted due to privacy concerns but linkable for research purposes and limited to use at the Health and Welfare Data Center (HWDC) only.

**Funding:** The authors thank for the research grand support from the Chang Gung Memorial Hospital, Linkou (CMRPG3J0201) to YTH, and the Ministry of Science and Technology (108-2410-H-182A-002) to YTH. We also acknowledge the support of the Maintenance Project of the Center for Big Data Analytics and Statistics (Grant CLRPG3D0048) at Chang Gung Memorial Hospital to SHC. The funders had no role in study design, data collection and analysis, decision to publish, or preparation of the manuscript.

**Competing interests:** The authors have declared that no competing interests exist.

the dengue-associated MACE should be taken into consideration when making public health management policies.

## Author summary

Dengue infection is the most rapidly spreading mosquito-borne viral illness worldwide and becomes a vivid threat to non-tropical countries. Previous research has documented that viral infection can increase risks of adverse cardiovascular events. There were sporadic reports about the association of cardiovascular events and dengue infection in the endemic tropical countries. Our study analyzed the risks for major adverse cardiovascular events, and found that acute myocardial infarction, stroke and heart failure were significantly higher in the immediate time period (within one week) after dengue infection, especially in patients with ≥60 years of age, female gender and severe admission dengue cases.

## Introduction

Viral infection can directly and indirectly affect the cardiovascular system, resulting in increased risks of major cardiovascular events [1,2]. Many viruses have been reported with case reports of major adverse cardiovascular events (MACEs), including influenza, severe acute respiratory syndrome virus (SARS-CoV), cytomegalovirus, Epstein-Barr virus, influenza, SARS-CoV-2, etc [3–7]. Various MACEs have been recognized, including acute myocardial infarction, stroke, heart failure, arrhythmia and myocarditis. In a milestone study, respiratory viral infection, especially influenza, had a significant association with acute myocardial infarction [8]. Moreover, vaccinations are very important in public health. Studies on the effect of influenza and pneumococcal vaccines have shown a 17 to 36% reduction in cardiovascular events [9,10].

In the modern era, globalization and climate change bring new public health challenges, such as the COVID-19 pandemic and emerging tropical viral infections, such as dengue virus. Dengue virus (DENV) infection is the most rapidly spreading mosquito-borne viral illness worldwide, and it is transmitted by *Aedes* mosquitoes, mainly of the species *Aedes aegypti*. The epidemiology of DENV has changed due to global warming [11,12]. An increased number of DENV infections has been reported in places with higher latitudes.

Large-scale studies on the association of MACEs and DENV infection are still lacking. Although there were some studies reporting on the association of MACEs and DENV infection, most of these reports were on youth and children in tropical endemic regions. Because the MACEs and immune response following viral infection can be completely different in the youth and in adult populations, these findings may not be generalized for populations in the regions where most adults are DENV immunity-naïve.

To understand the association of MACEs and DENV infection, we conducted this study using two nationwide databases that provided data on unbiased laboratory-confirmed DENV infection cases. The results will provide evidences for helping manage high-risk patients and make public health policy for dengue infection. The association between DENV and MACEs may provide a scientific clue for managing the potential MACEs in other viral infections such as COVID-19 in the future.

## Methods

### Ethics statement

This study was conducted in accordance with the Declaration of Helsinki and the Declaration of Taipei on ethical considerations regarding health databases by the World Medical Association. The study protocol was approved, and informed consent was exempted by the Institutional Review Board of Chang Gung Medical Foundation (IRB No: 201901517B0) as all data converted from the original NHI's claim records were anonymized and due to tight regulations of on-site analysis at HWDC.

### Data source

The data used in this study were retrieved from the National Health Insurance Research Database (NHIRD) and the Notifiable Disease Dataset of Confirmed Cases–Disease Prevention Database (NDDCC) under the Health and Welfare Data Center (HWDC) established by the Ministry of Health and Welfare (MoHW) of Taiwan. Both the NHIRD and the NDDCC are official databases, and they are managed by the National Health Insurance Administration and Centers for Disease Control of Taiwan (TwCDC), respectively. All databases were encrypted due to privacy concerns but linkable for research purposes and limited to use at HWDC only. Detailed information and description of the study data and specification of HWDC were reported in a previous article [13].

### Study design and populations

This is a population-based observation study. Data including all patients who had been initially diagnosed with dengue virus syndrome were collected from inpatient, emergency and outpatient department claims under the NHIRD between 2009 and 2015. We confirmed the diagnosis of DENV infection by obtaining their infection detection date (Id date) by linking to the NDDCC database. In Taiwan, all clinically suspected cases of DENV infection from physician offices, emergency departments, hospitals, long-term care facilities, and public health departments receive complete epidemiologic surveys and blood tests after being reported to the Taiwan Centers for Disease Control. The serology (IgM & IgG), real-time polymerase chain reaction (RT-PCR), and NS1 antigen were routinely examined for validation. Confirmed cases were all registered in the NDDCC database.

The hospitalization records for MACEs of laboratory-confirmed dengue cases were searched from the NHIRD inpatient dataset from 2008 to 2016. We included all subjects with MACEs one year before and after the DENV Id date. A schematic algorithm of the study is shown in Fig 1. If the MACEs were not related to DENV infection, then the incidence of MACEs for our selection subjects would presumably be distributed equally during the entire observation period. The first 7 and 14 days after the DENV Id date were defined as risk intervals, and the other periods were defined as control intervals. This study design was adapted from a previous study [8].

Due to potential concerns of bias, we added negative and positive control analyses. Onychomycosis, a nonviral and non-MACE-related infectious disease, was selected as our negative control. Infection with influenza virus served as our positive control. The analyses and research design for DENV infection and negative and positive controls were the same.

### Definition

The DENV patients were identified by *International Classification of Diseases, Ninth Revision, Clinical Modification* (ICD-9-CM) codes 061 and 065.4 and *International Classification of*

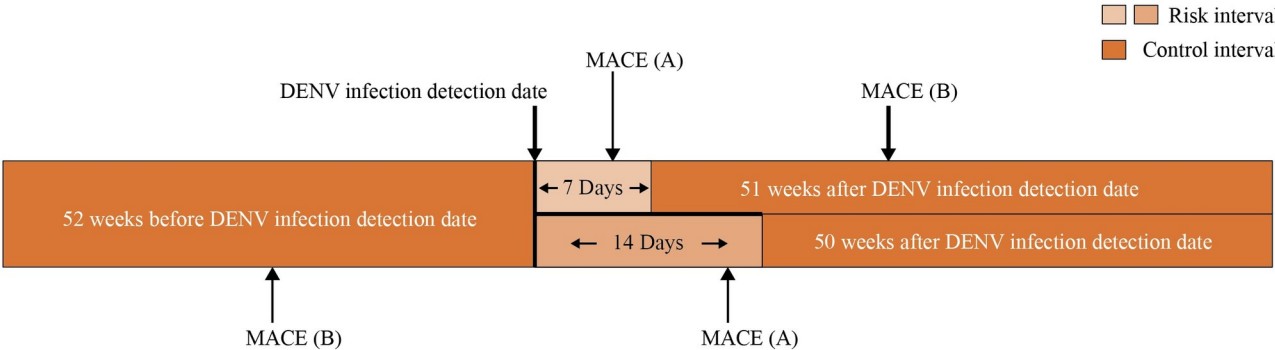

**Fig 1. Schematic of study design.** MACE (A) represents a person who is infected with DENV and is hospitalized for MACE (major adverse cardiovascular events) at any time during the 7-day risk interval (light-shaded areas) after exposure. Another MACE (B) represents a person infected with DENV who has an acute MACE during the control interval (dark-shaded areas). The study assessed the relative incidence of MACE during the risk interval as compared with the control interval. Note that the figure is not drawn to scale.

*Diseases*, *Tenth Revision* (ICD-10) code A90. MACEs, including hemorrhagic stroke (ICD-9-CM 430–432; ICD-10 I60-I62), ischemic stroke (ICD-9-CM 433–437; ICD-10 I63, I65-I67, G45, G46), acute myocardial infarction (AMI) (ICD-9-CM 410, 411; ICD-10 I20-I22, I24), heart failure (ICD-9-CM 428; ICD-10 I50), and thromboembolic stroke (ICD-9-CM 415.1, 453.4; ICD-10 I26, I80.2), were selected. The comorbidities that were major risk factors for MACEs included hypertension (ICD-9-CM 401–405; ICD-10 I10-I16), diabetes mellitus (ICD-9-CM 250; ICD-10 E08-E13), and disorder of lipid metabolism (ICD-9-CM 415.1, 453.4; ICD-10 I26, I80.2) and were also included in our analysis.

## Statistical analyses

We calculated the incidence rate ratio (IRR) and 95% confidence interval (C.I.) for MACEs per 1,000 during the risk interval and the control interval via Poisson regression. The model accounted for multiple DENV exposures and hospitalization episodes for MACEs per DENV patient during the observation period. We also evaluated the risks during days 8–14, days 15–21, and days 22–28 post-DENV infection. In addition, we performed analyses in subgroups defined according to age group (0–39 years, 40–59 years, and ≥60 years), sex, severity level (admission or nonadmission case), and stratification by each individual MACE item.

## Results

There were 79,037 patient consultations for dengue virus infection in Taiwan from Jan 1, 2009, to Dec 31, 2015. Of these, 65,906 patients were laboratory-confirmed to have DENV infection. The primary outcome of the incidence of MACEs within one year of dengue was observed in 1,247 patients (Fig 2). Summary data for DENV-infected patients with cardiovascular and neurovascular events are shown in Table 1. A total of 856 (68.6%) DENV patients with MACEs were hospitalized. Most of the vascular events (78.35%) were observed in patients older than 60 years. A 53% (929 vs. 606) increase in the number of hospitalizations due to cardiovascular and neurovascular events within 1 year after the diagnosis of DENV infection was found. The major MACEs came from ischemic stroke and heart failure. Because three patients' ages were missing, we included only 1,244 patients divided into 2,488 episodes in the subsequent analysis.

Table 2 shows the IRR for MACEs after laboratory-confirmed dengue infection. These events were highest within 7 days after having dengue fever, and the IRR of MACEs was 17.9

**Fig 2. Dengue virus infection cases included in the study.**

(95% C.I. 15.80–20.37). The IRR was still higher in the second week after DENV infection (IRR = 2.40, 95% C.I. 1.78–3.28), but there was no significant difference after the third (IRR = 1.38, 95% C.I. 0.93–2.05) and fourth (IRR = 1.10, 95% C.I. 0.71–1.71) week after DENV infection. A similar result was found in our positive control-influenza group. The first-week IRR for MACE was 15.76 (95% C.I. 14.88–16.70). On the contrary, the IRR was insignificant for onychomycosis (IRR = 0.91, 95% C.I. 0.79–1.05).

In the subgroup analyses, the major results were consistent with the main analysis but with discrepant effects. These results are shown in Table 3. The MACEs were highest in patients ≥60 years of age (IRR = 21.12, 95% C.I. 17.44–25.71) and lowest in patients 40–59 years old (IRR = 6.61, 95% C.I. 4.29–10.16). Females had a higher risk of MACEs than males (IRR 21.17 for females and 15.94 for males). The MACE risk was nearly six times higher among admission cases than among non-admission cases (IRR 24.39 vs 4.17). All of the IRRs of MACEs from each disease were over 10 during the first week after DENV infection, and the IRRs were 10.9

**Table 1. Baseline characteristics of patients with DENV infection who had a MACE within the observation period.**

| Characteristics | N | % |
|---|---:|---:|
| Age | | |
| 0–39 years | 30 | 2.41 |
| 40–59 years | 237 | 19.01 |
| ≥60 years | 977 | 78.35 |
| Missing | 3 | 0.24 |
| Sex | | |
| Male | 744 | 59.66 |
| Female | 503 | 40.34 |
| Hypertension | 839 | 67.28 |
| Diabetes mellitus | 480 | 38.49 |
| Dyslipidemia | 436 | 34.96 |
| Severity | | |
| Admission case | 856 | 68.64 |
| Non admission case | 391 | 31.36 |
| MACE before observation period | 606 | 48.60 |
| Hemorrhage stroke | 45 | 3.61 |
| Ischemic stroke | 228 | 18.28 |
| Acute myocardial infarction | 138 | 11.07 |
| Heart failure | 195 | 15.64 |
| MACE after observation period | 929 | 74.50 |
| Hemorrhage stroke | 94 | 7.54 |
| Ischemic stroke | 324 | 25.98 |
| Acute myocardial infarction | 174 | 13.95 |
| Heart failure | 337 | 27.02 |

MACE, major adverse cardiovascular events

for hemorrhagic stroke (95% C.I. 6.80–17.49), 13.53 for AMI (95% C.I. 10.13–18.06), 15.56 for ischemic stroke (95% C.I. 12.44–19.47), and 27.24 for heart failure (95% C.I. 22.67–32.73). The risks of ischemic stroke, hemorrhagic stroke and heart failure remained increased (IRR 4.33,

**Table 2. Incidence rate ratio for MACE after laboratory-confirmed dengue infection.**

| Variables | Incidence rate ratio (95% CI)[†] | | p value |
|---|---:|---|---:|
| Risk interval, days 1–28 | | | |
| Days 1–7 | 17.9 | (15.80–20.37) | < .0001 |
| Days 8–14 | 2.40 | (1.78–3.25) | < .0001 |
| Days 15–21 | 1.38 | (0.93–2.05) | 0.107 |
| Days 22–28 | 1.10 | (0.71–1.71) | 0.661 |
| Negative control | | | |
| Onychomycosis[‡] (n = 14,760) | 0.91 | (0.79–1.05) | 0.184 |
| Positive control | | | |
| Influenza[‡] (n = 4,553) | 15.76 | (14.88–16.70) | < .0001 |

MACE, major adverse cardiac event; CI, confidence interval

[†] Estimated by Poisson regression with adjusted baseline characteristics

[‡] Risk interval: days 1–7

**Table 3. Subgroup analyses comparing incidence rate ratios for MACE after dengue infection.**

| Variables | Day 1–7 | | | Day 8–14 | | |
|---|---|---|---|---|---|---|
| | N | IRR (95% CI)[†] | p value | N | IRR (95% CI)[†] | p value |
| Age | | | | | | |
| 0–39 years | 60 | 16.66 (6.48–42.84) | < .0001 | 60 | 3.00 (0.40–21.54) | 0.286 |
| 40–59 years | 474 | 6.61 (4.29–10.16) | < .0001 | 474 | 2.60 (1.34–5.05) | 0.005 |
| ≥60 years | 1,954 | 21.12 (18.45–24.18) | < .0001 | 1,954 | 2.34 (1.66–3.30) | < .0001 |
| Sex | | | | | | |
| Male | 1,488 | 15.94 (13.48–18.86) | < .0001 | 1,488 | 2.16 (1.44–3.23) | 0.000 |
| Female | 1,006 | 21.17 (17.44–25.71) | < .0001 | 1,006 | 2.79 (1.77–4.40) | < .0001 |
| Severity | | | | | | |
| Admission case | 1,712 | 24.39 (21.30–27.92) | < .0001 | 1,712 | 2.42 (1.69–3.45) | < .0001 |
| Non admission case | 782 | 4.17 (2.70–6.45) | < .0001 | 782 | 2.34 (1.32–4.15) | 0.004 |
| With disease hospitalization | | | | | | |
| Hemorrhage stroke | 234 | 10.90 (6.80–17.49) | < .0001 | 234 | 4.33 (2.13–8.78) | < .0001 |
| Ischemic stroke | 952 | 15.56 (12.44–19.47) | < .0001 | 952 | 3.17 (2.03–4.94) | < .0001 |
| Acute myocardial infarction | 544 | 13.53 (10.13–18.06) | < .0001 | 544 | 1.16 (0.48–2.81) | 0.737 |
| Heart failure | 810 | 27.24 (22.67–32.73) | < .0001 | 810 | 2.45 (1.49–4.03) | <0.001 |

MACE, major adverse cardiac event; IRR, incidence rate ratios; CI, confidence interval

[†] Estimated by Poisson regression with adjusted baseline characteristics

3.17, and 2.45, respectively), but insignificance was observed among AMI patients in the second week of DENV infection.

## Discussion

Our study found that the risk of MACEs was increased in patients within the early two weeks of DENV infection. The possible mechanisms for increased MACEs following DENV infection are unclear and complex. Many proposed viral, host response and immune mechanisms are known to be involved in the cardiac and vascular manifestations of DENV infections.

Viral infection can directly and indirectly augment cardiovascular risks [1,2]. DENV can cause direct infection of the cardiovascular system. DENV-induced myocarditis and myopathy have been reported, as DENV antigen was seen in cardiac tissues, including cardiomyocytes, myocardial interstitial cells, and endothelial cells [14]. DENV-infected vascular endothelium may also be responsible for both endothelial leakage and a longer effect on endothelial function. Children and adolescents with a history of dengue hemorrhagic fever have been reported to have an increased carotid intima-media thickness [15].

The host inflammatory response to infection often results in the release of proinflammatory cytokines and the activation of platelets, leukocytes, and endothelial cells that can activate procoagulant pathways while inhibiting anticoagulant pathways [16]. Vasoactive and proinflammatory cytokines increase capillary leakage, and endothelial dysfunction may also cause myocardial dysfunction. Altered intracellular calcium homeostasis potentially causes electrical abnormalities that cause arrythmia or desynchronized myocardial movement and eventually cause cardiovascular events [17].

Vascular thrombosis is a major complication of infection and may be related to the pathogenesis of DENV-related MACEs [3,4]. Case reports of virus-associated thrombosis have been reported in many kinds of viruses, such as SARS-CoV-2, cytomegalovirus, Epstein-Barr virus, influenza, etc [3–7]. DENV infection is known for its hemorrhagic features but has also been

reported in some cases of major thromboembolic events, including ischemic stroke, deep vein thrombosis, pulmonary embolism, central vein thrombosis, and mesenteric vein thrombosis [7,18–20].

Platelet dysfunction may play a crucial role in DENV-associated ischemic stroke and myocardial infarctions. Although thrombocytopenia is a common presentation of DENV infection, rebound thrombocytosis has also been reported [21]. Thrombocytosis may increase the risk of thromboembolic events, such as myocardial infarction, stroke, venothromboemboli and congestive heart failure.

Risks for cardiovascular events such as myocardial infarction or stroke have been reported to increase within 180 days after bacteremia and influenza infection [22]. In our study, DENV-associated thrombotic events mostly occurred within the first 2 weeks of illness and were highest within the first 7 days. Previous reports on DENV-associated thrombosis showed similar findings [19,23]. Some patients with no risk factors for thrombosis, such as smoking, obesity or use of contraceptives, have thrombotic events during the first 5 days of DENV [19]. Most reported cases of DENV-associated thrombosis were seen in patients 31–60 years of age [7,19,23,24]. Our patients were mostly male and more than 40 years of age and had a medical history of hypertension, diabetes mellitus and hyperlipidemia (67.28%, 38.49%, and 34.96%, respectively). Li et al reported that the incidence of stroke in DENV infection was higher in male patients with comorbidities. They concluded that these factors are strong risk factors that may mask the effect of dengue fever on the occurrence of stroke [20].

In addition to thromboembolic events, heart failure was significantly increased. DENV can affect heart function via different mechanisms. The virus is taken in by macrophages, causing the activation of T cells and subsequent release vasoactive and pro-inflammatory cytokines. The release of these cytokines leads to capillary leakage and possible myocardial damage [17]. The interaction between NS1 protein and vascular endothelial glycocalyx layer is thought to increase capillary permeability. The resulting plasma leakage can result to myocardial interstitial edema and cause cardiac dysfunction by reducing preload and alteration of coronary microcirculation. Changes in intracellular calcium homeostasis in DENV-infected myotubes could lead to contractile dysfunction and ultimately to decompensation of heart failure [17].

DENV-associated MACEs may be underreported because many patients are asymptomatic or mildly symptomatic. Taking abnormal cardiovascular rhythm as an example, mild arrhythmia is common and frequently ignored. A study in Brazil demonstrated that cardiovascular manifestations were common (19.7%), as arrhythmias (sinus bradycardia [13.8%], atrial [4.9%] and ventricular [4.0%] extrasystoles) were frequently observed if electrocardiogram was routinely performed for DENV-infected patients [25]. Five percent of DENV patients had prolongation of the QTc interval [26]. In a DENV outbreak of 2005 in Sri Lanka, ECG abnormalities were found in 62.5% of patients (the median age: 34, ranging 12 to 76 years old), and 80% of them developed transient hypotension and tachycardia-bradycardia syndrome suggestive of transient severe cardiac dysfunction [27]. These may be associated with significant morbidity and higher mortality if they affect specific high-risk patient groups, such as the elderly or patients with underlying cardiac conditions.

## Limitations

There are some limitations in this study. At higher latitudes, cases of DENV infection are usually clustered during summer, which may cause bias because the distribution of MACE incidence is affected by the season. However, there was a small seasonal variation in endemic DENV in Taiwan. Moreover, based on our self-controlled design including one year before and after, the effect of seasonal variation should be eliminated.

Laboratory data, such as platelet count and coagulation status, were lacking in the database. Information about smoking, diet, body mass index and daily activity was not recorded in the NHIRD. Ideally, similar viral infections should be chosen as controls. Patients infected with influenza served as positive controls as influenza has been known to increase the risk of MACEs. However, it was not possible to choose a viral disease that is a CDC-reportable infectious disease and has been proven not to cause MACEs. Thus, onychomycosis, a common localized dermatophyte infection without known systemic infection, was chosen as the negative control. In this study, we did not analyze the different serotypes of DENV, as in the past endemic of 2014–2015 almost all cases were DENV-1 and DENV-2 infections.

Finally, the majority of exposure dates were based on the date of disease onset. It is possible that the date of disease onset was inaccurate in some cases as a result of patient recall bias and error in data entry. However, the Communicable Disease Control Act of Taiwan demands physicians to formally report a case of dengue infection to the government authority within 24 hours if the diagnosis of dengue fever is suspected. Although there is probably an error of a few days in some cases, the risk period was measured in weeks in this study, hence the discrepancy is not expected to significantly affect the results.

## Conclusion

In conclusion, our study found that MACEs are associated with acute DENV infection, which is important because cardiovascular events triggered by dengue virus are potentially preventable by early recognition, public health management and vaccination. The risk for MACEs, including AMI, heart failure and stroke, is significantly higher in the immediate time period after dengue virus infection. Patients with dengue fever should be carefully monitored during the acute phase of disease to ensure early recognition of symptoms of impending AMI, stroke or heart failure. Influenza vaccines have been proven to reduce cardiovascular events, and the benefit is even greater in the population with known coronary artery disease [9,10]. Whether dengue virus vaccination reduces the risk of MACEs in patients with a high risk of cardiovascular disease is an intriguing question.

## Supporting information

**S1 Table. Baseline characteristics of patients with onychomycosis and influenza infection who had a MACE within the observation period**
(DOCX)

## Acknowledgments

The authors thank the Center for Big Data Analytics and Statistics at Chang Gung Memorial Hospital for the statistical assistance.

## Author Contributions

**Conceptualization:** Kai-Che Wei, Yu-Tung Huang.

**Data curation:** Kai-Che Wei, Chia-Ling Wu, Shang-Hung Chang, Yu-Tung Huang.

**Formal analysis:** Chia-Ling Wu, Yu-Tung Huang.

**Funding acquisition:** Shang-Hung Chang, Yu-Tung Huang.

**Investigation:** Kai-Che Wei, Cheng-Len Sy, Wen-Hwa Wang, Yu-Tung Huang.

**Methodology:** Kai-Che Wei, Yu-Tung Huang.

**Project administration:** Chia-Ling Wu.

**Resources:** Shang-Hung Chang, Yu-Tung Huang.

**Supervision:** Shang-Hung Chang, Yu-Tung Huang.

**Validation:** Kai-Che Wei, Shang-Hung Chang, Yu-Tung Huang.

**Visualization:** Chia-Ling Wu.

**Writing – original draft:** Kai-Che Wei, Cheng-Len Sy, Wen-Hwa Wang.

**Writing – review & editing:** Shang-Hung Chang, Yu-Tung Huang.

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
