## [Decision Letter · Decision Letter 0]

4 Nov 2021

Dear Associate Research Fellow Huang,

Thank you very much for submitting your manuscript "Major acute cardiovascular events after dengue infection– A population-based observational study" for consideration at PLOS Neglected Tropical Diseases. As with all papers reviewed by the journal, your manuscript was reviewed by members of the editorial board and by several independent reviewers. The reviewers appreciated the attention to an important topic. Based on the reviews, we are likely to accept this manuscript for publication, providing that you modify the manuscript according to the review recommendations. 

Sincerely,

Husain Poonawala

Associate Editor

Paul Newton

Deputy Editor

Reviewer's Responses to Questions

**Key Review Criteria Required for Acceptance?**

**Methods**

-Are the objectives of the study clearly articulated with a clear testable hypothesis stated?

-Is the study design appropriate to address the stated objectives?

-Is the population clearly described and appropriate for the hypothesis being tested?

-Is the sample size sufficient to ensure adequate power to address the hypothesis being tested?

-Were correct statistical analysis used to support conclusions?

-Are there concerns about ethical or regulatory requirements being met?

Reviewer #1: all are yes.

Reviewer #2: Yes.

**Results**

-Does the analysis presented match the analysis plan?

-Are the results clearly and completely presented?

-Are the figures (Tables, Images) of sufficient quality for clarity?

Reviewer #1: all are yes.

Reviewer #2: The analysis presented match she analysis plan and the results are clearly and completely presented. However, a table describing the demographic characteristics of all the groups included in the study, including the individuals who had influenza and onychomycosis, should be added.

**Conclusions**

-Are the conclusions supported by the data presented?

-Are the limitations of analysis clearly described?

-Do the authors discuss how these data can be helpful to advance our understanding of the topic under study?

-Is public health relevance addressed?

Reviewer #1: all are yes.

Reviewer #2: Yes,

**Editorial and Data Presentation Modifications?**

Reviewer #1: age group (0-40 years, “40-60 years” [41-60 years?], and >60 years).

Reviewer #2: (No Response)

**Summary and General Comments**

Reviewer #1: The current study was aimed to evaluate the association between Dengue virus (DENV) infection and major adverse cardiovascular events (MACEs, including acute myocardial infarction [AMI], heart failure and stroke). This is a population-based observation study. All laboratory confirmed dengue cases in Taiwan during 2009 and 2015 were included by Center for Disease Control (CDC) notifiable database. The primary outcome of the incidence of MACEs within one year of dengue was observed in 1,247 patients. They found that AMI, stroke and heart failure are significantly higher in the immediate time period (within one week) after dengue infection, especially in patients with >60 years of age, female gender and severe admission dengue cases. The authors concluded that cardiovascular events triggered by dengue virus are potentially preventable. Patients with dengue fever should be carefully monitored during the acute phase of disease to ensure early recognition of symptoms of AMI, stroke or heart failure. Whether dengue virus vaccination reduces the risk of MACEs in patients with a high risk of cardiovascular disease is an intriguing question.

This is an interesting study with some novel findings and clinical implications. The study was well-conducted. The conclusions were supported by the results. The manuscript was well-written.

Some points concerned:

1. On page 3, line 6: …. the “dengue associated of MACE” (“dengue-associated MACE” ?) should be taken into……

2. On page 3, line 10: …. becomes a vivid “thread” (“threat”?) to non-tropical countries…..

3. On page 4, line 7: …stroke, heart failure, arrhythmia and “cardiomyositis” (“myocarditis”?)…

4. On page 9, line 6-7, (also in Tables 1, 3): ... we performed analyses in subgroups defined according to age group (0-40 years, “40-60 years” [41-60 years?], and >60 years), sex,…..

5. On page 14, line 10-11: …The risks of “ischemic stroke hemorrhagic stroke and heart failure” (“hemorrhagic stroke, ischemic stroke and heart failure”?) remained increased (IRR 4.33, 3.17, and 2.45, respectively),…..

6. On page 19, line 4-5: …contractility and eventually leading (“to”?) decompensation……

7. On page 19, line 10-11: …were frequent if “routine” electrocardiogram was “routinely” (redundant? Deleted?) performed for DENV-infected…..

8. On page 19, line 1 from the bottom: …if they attack in some specific patient groups, such as the “elder” (“elderly”?). 

9. On page 20, line 2-3: …At higher “altitudes” (latitude”?), cases of DENV infection are usually clustered during summer,……

Reviewer #2: The manuscript entitled “Major acute cardiovascular events after dengue infection– A population-based observational study” by Wei et al. is well written and reports interesting results. This study aimed to perform a large-scale evaluation of the association between Dengue virus infection and major adverse cardiovascular events. They observed significantly increased incidence rate ratio for major cardiovascular events (hemorrhagic stroke, ischemic stroke, acute myocardial infarction, and heart failure) mainly within the first week after the onset of Dengue virus infection. The study design is adequate and the results support its conclusions. However, some adjustments should be performed in the manuscript:

1- The text excerpt “DENV is taken up into macrophages…DENV-infected myotubes” (lines 252-259) was literally copied from the study cited by the reference 17, which is considered as plagiarism. Therefore, the aforementioned excerpt should be completely rewritten and all the manuscript should undergo a careful plagiarism check.

2- The reference 16 does not support the information from lines 220-222. Another study that supports the data from that excerpt should be chosen to replace the reference 16.

3- I suggest the inclusion of a table describing the demographic characteristics of all the groups included in the study, including the individuals who had influenza and onychomycosis.

Below are some language errors that should be eliminated:

1- Line 73: The word “Aedes” should be written in italics.

2- Line 75: The word “attitudes” should be replaced with “altitudes”.

3- Line 110: The word “and” should be added before the word “NS1”.

4- Line 200: A comma should be added between the words “ischemic stroke” and “hemorrhagic stroke”.

PLOS authors have the option to publish the peer review history of their article (what does this mean?). If published, this will include your full peer review and any attached files.

Reviewer #1: No

Reviewer #2: No

Figure Files:

Data Requirements:

Reproducibility:

References

---

## [Decision Letter · Decision Letter 1]

3 Jan 2022

Dear Associate Research Fellow Huang,

We are pleased to inform you that your manuscript 'Major acute cardiovascular events after dengue infection– A population-based observational study' has been provisionally accepted for publication in PLOS Neglected Tropical Diseases.

Best regards,

Husain Poonawala

Associate Editor

Paul Newton

Deputy Editor

Reviewer's Responses to Questions

**Key Review Criteria Required for Acceptance?**

**Methods**

-Are the objectives of the study clearly articulated with a clear testable hypothesis stated?

-Is the study design appropriate to address the stated objectives?

-Is the population clearly described and appropriate for the hypothesis being tested?

-Is the sample size sufficient to ensure adequate power to address the hypothesis being tested?

-Were correct statistical analysis used to support conclusions?

-Are there concerns about ethical or regulatory requirements being met?

Reviewer #1: All are yes.

Reviewer #2: (No Response)

**Results**

-Does the analysis presented match the analysis plan?

-Are the results clearly and completely presented?

-Are the figures (Tables, Images) of sufficient quality for clarity?

Reviewer #1: All are yes.

Reviewer #2: (No Response)

**Conclusions**

-Are the conclusions supported by the data presented?

-Are the limitations of analysis clearly described?

-Do the authors discuss how these data can be helpful to advance our understanding of the topic under study?

-Is public health relevance addressed?

Reviewer #1: All are yes.

Reviewer #2: (No Response)

**Editorial and Data Presentation Modifications?**

Reviewer #1: None.

Reviewer #2: (No Response)

**Summary and General Comments**

Reviewer #1: The manuscript has been revised satisfactorily according to the reviewer's comments and suggestions.

Reviewer #2: The authors have edited the manuscript according the suggestions from peer review.

PLOS authors have the option to publish the peer review history of their article (what does this mean?). If published, this will include your full peer review and any attached files.

Reviewer #1: No

Reviewer #2: No

---

## [Editor Report · Acceptance letter]

1 Feb 2022

Dear Associate Research Fellow Huang,

We are delighted to inform you that your manuscript, "Major acute cardiovascular events after dengue infection– A population-based observational study," has been formally accepted for publication in PLOS Neglected Tropical Diseases.

Best regards,

Shaden Kamhawi

co-Editor-in-Chief

Paul Brindley

co-Editor-in-Chief
